# Phytosterol and Polyphenol Contents and Quinoa Leave Yields Variation in Relationships to Variety, Density and Harvesting Date

Vasile Stoleru [1,*], Maricel Vitanescu [1], Gabriel-Ciprian Teliban [1], Alexandru Cojocaru [1], Laurian Vlase [2], Ana-Maria Gheldiu [3], Ionel Mangalagiu [4], Dorina Amăriucăi-Mantu [4], Marian Burducea [4], Valtcho Zheljazkov [5] and Oana-Raluca Rusu [6]

1   Department of Horticulture, "Ion Ionescu de la Brad" University of Life Sciences, 3 M. Sadoveanu, 700440 Iasi, Romania
2   Department of Pharmaceutical Technology and Bio-pharmaceutics, "Iuliu Hatieganu" University of Medicine and Pharmacy, 12 Ion Creanga, 400347 Cluj-Napoca, Romania
3   Department of Pharmaceutical Botany, Faculty of Pharmacy, "Iuliu Hatieganu" University of Medicine and Pharmacy, 23 Gheorghe Marinescu Street, 400337 Cluj-Napoca, Romania
4   Departments of Biochemistry and SCDAEA, "Alexandru Ioan Cuza" University of Iasi, Carol I Boulevard 20A, 700506 Iasi, Romania
5   Department of Crop and Soil Science, Oregon State University, 3050 SW Campus Way, 109 Crop Science Building, Corvallis, OR 97331, USA
6   Department of Public Health, "Ion Ionescu de la Brad" University of Life Sciences, 6 M. Sadoveanu, 700449 Iasi, Romania
*   Correspondence: vstoleru@uaiasi.com; Tel.: +40-743-180-275

**Abstract:** Quinoa is an Andean grain known for its gluten-free grains, which are used as a functional food. The aim of this research was to study the possibility of introducing quinoa as a vegetable crop grown for young leaves as a source of polyphenols and phytosterols. To achieve this goal, a field experiment was performed with three quinoa cultivars (Titicaca, Puno, and Vikinga) grown in a split plot design. The experimental factors included three densities (160, 320, and 760 plants·m$^{-2}$) and two harvest dates (52 and 62 days after sowing (DAS)). The content of phytosterols (ergosterol, stigmasterol, β-sitosterol and campesterol) and polyphenols (ferulic acid, isoquercitrin and rutozid) in quinoa leaves were determined by HPLC method. The phytosterol content varied within the limits: ergosterol 0–7.62 µg·100 g$^{-1}$ dw, stigmasterol 79.9–175.3 µg·100 g$^{-1}$ dw, β-sitosterol 425.7–623.1 µg·100 g$^{-1}$ dw and campesterol 0–5.25 µg·100 g$^{-1}$ dw. Ferulic acid varied greatly from 38.0 to 63.3 µg·g$^{-1}$ dw, isoquercitrin ranged from 63 to 101.6 µg·g$^{-1}$ dw and rutozid varied widely from 32.9 to 162.8 µg·g$^{-1}$ dw. The total phytosterols and the total phenolic compounds decreased with decreasing plant number density and DAS. This research demonstrated that young quinoa leaves are a good source of phytosterols and phenolics for human consumption.

**Keywords:** *Chenopodium quinoa*; cultivation technology; young leaves; bioactive compounds

## 1. Introduction

Quinoa (*Chenopodium quinoa* Willd) has recently gained great popularity due to its nutritional properties such as a rich content of protein, lipids and fiber. Moreover, quinoa is resistant to environmental stresses such as low temperatures, drought and salinity, which makes it an adaptable crop in different regions of the world [1]. Quinoa seeds have high quality protein content due to the presence of all essential amino acids, while being rich in lysine and methionine which are deficient in cereals [2]. Although part of the Amaranthaceae family, quinoa is considered a pseudocereal because its seeds are morphologically an "achene" as the monocotiledons [1]. Quinoa is considered a functional food with multiple uses [3]. The main quinoa producing countries are Peru, Bolivia,

Ecuador, Argentina, Chile, and Colombia; recently, it has been introduced in Europe, North America, and Africa [4]. The importance of quinoa as a sustainable crop for food systems and food security was recognized worldwide by the declaration of 2013 as the International Year of Quinoa by FAO [5]. Although quinoa is generally grown for its seeds, some reports have highlighted the nutritional quality of its leaves. Quinoa leaves are a rich source of nutrients (proteins 27–30 g·kg$^{-1}$, Na 289 mg·100 g$^{-1}$, ash 3.3%, fiber 1.9%, nitrates 0.4%) and other bioactive compounds (vitamin E 2.9 mg α-TE·100 g$^{-1}$ and vitamin C 1.2–2.3 g·kg$^{-1}$), and can be consumed like spinach or in salads [2]. In comparison with amaranth and spinach leaves, it was shown that the content of proteins and essential amino acids in quinoa leaves was higher, while that of carbohydrates was lower [6]. An important aspect regarding the nutritional value of quinoa leaves is that the content of anti-nutritional factors such as phytic acid, oxalates, saponins, the trypsin inhibitor or α-amylase is low (0.03–0.06 g·100 g$^{-1}$ fw; 0.11–0.25 g·100 g$^{-1}$ fw; 0.07–0.15 g·100 g$^{-1}$ fw; 0.34–0.62 TUI·mg$^{-1}$ fw; and 0.13–0.31 g·100 g$^{-1}$ fw, respectively) [7].

Recent research has shown that quinoa leaves have antioxidant and anticancer properties due to their phenolic acid content with high bioaccessibility and bioavailability [8]. Phenolic compounds are products of secondary metabolism and have the role of protection against UV radiation and herbivores, but also to attract insects to perform pollination, or act as signaling molecules in physiological processes [9]. For this reason, it is recommended to consume plants with a rich content of phenolic compounds, due to the antioxidant properties that can prevent the appearance of diseases such as diabetes, cardiovascular disease, cancer, etc. [10]. Złotek et al., 2019 [11], recommend the consumption of quinoa sprotus due to a richer content of phenolic compounds, such as vanillic, p-coumaric and ferulic compared with quinoa seeds. Quinoa leaves have a rich content of phenols and flavonoids (131.8 ± 10.3 mg·100 g$^{-1}$ and 62.07 ± 5.1 mg·100 g$^{-1}$, respectively), the main compounds being gallic acid, kaempferol and catechin [12]. Tang et al., 2014 [13], analyzed the composition of leaves from six varieties of quinoa to determine their nutritional value, showing that the leaves contain a favorable ratio of fatty acids, ω-3/ω-6, α- and β-tocopherols, Pro-vitamin A, α- and β-carotenes, lutein and violaxanthin.

Another category of bioactive compounds with an important role in plant physiology is the class of phytosterols [14]. Over 250 phytosterols are known, while the most common being 4-desmethylsterols, sitosterol, stigmasterol and campesterol [15]. Phytosterols act as a structural component of membranes and hormone precursors. These compounds are analogues of cholesterol in animals and have anticancer, antidiabetic properties, and can prevent cardiovascular disease [16]. To date, the phytosterol content of quinoa leaves has not been investigated.

Based on the information reported above, this study aimed to evaluate the possibility of introducing quinoa cultivation and consumption of alternative nutritious leaves. Due to its adaptability to various pedological and climatic conditions, quinoa has rich genetic diversity. Moreover, by selecting the cultivar and the cultivation technology adapted to the environmental conditions, there is the possibility of obtaining crops with improved production and biological characteristics [17,18].

In this regard, the aim of the study was to evaluate the effect of crop management on leaf yield and phenolic and phytosterols content in quinoa cultivars. This study reports for the first time the content of phytosterols in young quinoa leaves.

## 2. Materials and Methods

### 2.1. Experimental Plots Design

The research was carried out at an experimental station in Cudalbi, Galati County. The area is characterized by average monthly temperatures of −5.3–22.3 °C, relative humidity between 59 and 84% and insolation ranging from 22 to 308.7 kWh·m$^{-2}$ (Table 1).

**Table 1.** Climatic conditions of experimental station, Galati (Romania)—2017.

| Months | Average Temperature (°C) | Relative Humidity (%) | Insolation (kWh·m$^{-2}$) |
|---|---|---|---|
| January | −5.3 | 77 | 22.9 |
| February | −0.3 | 82 | 44.2 |
| March | 7.6 | 70 | 124.3 |
| April | 9.1 | 63 | 171.0 |
| May | 16.1 | 62 | 261.1 |
| June | 21.1 | 65 | 278.5 |
| July | 21.6 | 67 | 279.0 |
| August | 22.3 | 59 | 308.7 |
| September | 17.8 | 60 | 205.3 |
| October | 10.5 | 74 | 150.8 |
| November | 6.1 | 84 | 35.4 |
| December | 0.4 | 82 | 30.5 |

The research was organized in a split plot design with three replications (Table 2). Experimental factors: three quinoa cultivars (Titicaca—T, Puno—P, and Vikinga—V), three crop densities (D1—160 plants·m$^{-2}$, D2—320 plants·m$^{-2}$, and D3—760 plants·m$^{-2}$) and two harvest times (13 November at 52 days after sowing DAS—H1 and 23 November at 62 DAS—H2).

**Table 2.** Experimental plot design.

| Variant | Cultivar | Density (Plants·m$^{-2}$) | Harvest Date |
|---|---|---|---|
| V1 | | 160 | |
| V2 | Titicaca | 320 | |
| V3 | | 760 | |
| V4 | | 160 | 13 November |
| V5 | Puno | 320 | (52 DAS) |
| V6 | | 760 | |
| V7 | | 160 | |
| V8 | Vikinga | 320 | |
| V9 | | 160 | |
| V10 | | 160 | |
| V11 | Titicaca | 320 | |
| V12 | | 760 | |
| V13 | | 160 | 23 November |
| V14 | Puno | 320 | (62 DAS) |
| V15 | | 760 | |
| V16 | | 160 | |
| V17 | Vikinga | 320 | |
| V18 | | 760 | |

The Titicaca cultivar with a cultivation density of 160 plants·m$^{-2}$, harvested at 52 DAS represents the control of the experience (V1).

Sowing time was the first of September. Harvesting was done at growth stage four according to the BBCH scale (Development of harvestable vegetative parts) before flowering. The seeds of the three quinoa cultivars were obtained from Quinoa Quality ApS (Denmark).

The quinoa crop was established on a chernozem chambic soil with medium texture. The cultivation was carried out in accordance with the technological norms described by Stoleru et al., 2014 [19].

### 2.2. Morphological Parameters and Yield of Young Quinoa Plants

After harvesting, the following parameters were determined: the leaves number, leaf area ($mm^2$), fresh leaf mass (g), and the fresh yield ($t \cdot ha^{-1}$). Leaf area was measured with LI-3100C Area Meter, LI-COR, (Lincoln, NE, USA).

### 2.3. Extracts of Quinoa

The content of different phytosterols and phenolic compounds from *Chenopodium quinoa* Willd. was determined in 70% hydroalcoholic solutions. The extractions were performed under ultrasound, using a Sonics reactor VCX-130, (Newtown, CT, USA) with a frequency of 20 KHz and a nominal power of 130 W. In this respect, for each sample, 0.2 g of quinoa chopped leaves was mixed with 25 mL of hydroalcoholic solutions, and subjected to ultrasound irradiation for five minutes, at room temperature (22 °C). The conditions used for the extractions consisted of applying a pulse of irradiation (5 s pulse/5 s pause) using 50% of the full power of the generator [20].

### 2.4. HPLC-MS Analysis of Phytosterolic Compounds

It was used the method described by Vlase et al., 2013 [20]. The quantitative content of the phenolic or sterolic compounds was estimated using HPLC-MS (high performance liquid chromatography coupled with mass spectrometry) techniques. In the chromatographic conditions, the phytosterols and phenols were eluted with the following retention times and $m/z$: ergosterol 3.2 min, 396 $m/z$; stigmasterol 4.9 min, 412 $m/z$; campesterol 4.9 min, 400 $m/z$; β-sitosterol 5.7 min, 414 $m/z$ for; 12.8 min, ferulic acid 193 $m/z$; isoquercitrin 20.29 min, 463 $m/z$; rutoside 20.76 min, 609 $m/z$. Because in the ionization conditions all sterols have lost a water molecule, the ions detected are always in the form $[M - H_2O + H]^+$.

### 2.5. Statistical Analysis

Results were expressed as means with standard errors. To assess significance differences between treatments, means were compared by one-way analyses of variances (ANOVA). When results were statistically significant, Tukey post hoc multiple comparisons test was used [21]. The mean difference was considered to be significant at $p < 0.05$ level. The software used for the statistical analyses was IBM SPSS v20.

## 3. Results

### 3.1. Fresh Yield and Morphology

In this study, three varieties of quinoa (Titicaca, Puno and Vikinga) were grown to obtain young plants for fresh consumption. The plants were cultivated using three different densities (D1—160 plants·$m^{-2}$, D2—320 plants·$m^{-2}$, and D3—760 plants·$m^{-2}$) and were harvested after 52 days and 62 days after sowing (DAS), respectively. The number of days until harvest was set so that the plants reached the vegetation stage just before the inflorescence appeared (stage four according to the BBCH scale—development of harvestable vegetative parts). The influence of variety, sowing density and harvest date on plant morphology and quinoa production is presented in Table 3. Morphological parameters varied depending on the variety, so the highest number of leaves was obtained at Puno, the leaf area was the largest at Vikinga and the largest mass of fresh leaves at Titicaca. The yield was higher in Titicaca by 53% compared to Puno and by 22% compared to Vikinga. The optimal density for morphological parameters is 320 plants·$m^{-2}$, while for production the optimal density is 760 plants·$m^{-2}$. In terms of harvest time, all morphological parameters, including production, were higher at 62 DAS.

**Table 3.** Cultivar, plant density and harvesting time effect on quinoa leaf cultivars.

| Treatment | Leaves Number | Leaf Area (mm$^2$) | Fresh Leaf Mass (g) | Fresh Yield (t·ha$^{-1}$) |
|---|---|---|---|---|
| Cultivar | | | | |
| Titicaca | 16.34 ± 0.56 a | 103.52 ± 7.15 c | 8.76 ± 0.41 a | 32.42 ± 3.40 a |
| Puno | 16.39 ± 0.39 a | 107.26 ± 8.50 b | 5.44 ± 0.33 c | 21.11 ± 2.79 c |
| Vikinga | 13.95 ± 0.44 b | 117.92 ± 10.62 a | 6.39 ± 0.32 b | 26.47 ± 3.81 b |
| Crop density | | | | |
| D1 160 plants·m$^{-2}$ | 14.70 ± 0.43 c | 84.46 ± 4.25 c | 7.26 ± 0.57 b | 11.62 ± 0.92 c |
| D2 320 plants·m$^{-2}$ | 16.73 ± 0.60 a | 141.14 ± 5.75 a | 7.62 ± 0.46 a | 24.39 ± 1.49 b |
| D3 760 plants·m$^{-2}$ | 15.25 ± 0.46 b | 103.09 ± 9.66 b | 5.71 ± 0.24 c | 43.98 ± 1.82 a |
| Harvest date | * | * | * | ns |
| 52 DAS | 14.29 ± 0.31 | 91.04 ± 5.73 | 6.13 ± 0.42 | 24.00 ± 2.69 |
| 62 DAS | 16.83 ± 0.41 | 128.09 ± 6.82 | 7.59 ± 0.31 | 29.33 ± 2.93 |

Values associated with different letters are significantly different according to Tukey's test at $p < 0.05$. For Harvest time ANOVA test was performed (* $p < 0.05$, ns—not significant). DAS—days after sowing.

The influence of the interaction between the studied factors (variety, sowing density and harvest date) on quinoa production is presented in Table 4. A heterogeneity of morphological parameters was found, with the largest number of leaves at T × D2 × H2, the largest leaf area at V × D3 × H2, the largest fresh mass at control variant T × D1 × H1, and the largest production at V × D3 × H2.

**Table 4.** Cultivar, plant density and harvesting time interaction effect on quinoa leaf cultivars.

| Interaction | Leaves Number | Leaf Area (mm$^2$) | Fresh Leaf Mass (g) | Fresh Yield (t·ha$^{-1}$) |
|---|---|---|---|---|
| T × D1 × H1 | 13.78 ± 0.15 fgh | 74.29 ± 0.12 n | 10.62 ± 0.10 a | 17.00 ± 0.16 h |
| T × D2 × H1 | 14.77 ± 0.43 defg | 108.46 ± 0.37 gh | 8.85 ± 0.05 c | 28.33 ± 0.15 ef |
| T × D3 × H1 | 15.00 ± 0.47 cdefg | 64.81 ± 0.13 o | 6.36 ± 0.03 e | 48.97 ± 0.22 b |
| P × D1 × H1 | 14.61 ± 0.31 efg | 56.14 ± 0.61 p | 4.43 ± 0.05 g | 7.13 ± 0.08 k |
| P × D2 × H1 | 15.61 ± 0.31 cde | 157.59 ± 0.30 c | 4.62 ± 0.02 g | 14.77 ± 0.05 i |
| P × D3 × H1 | 16.55 ± 0.29 bc | 94.29 ± 0.15 j | 3.92 ± 0.04 h | 30.24 ± 0.34 e |
| V × D1 × H1 | 12.50 ± 0.29 hi | 77.43 ± 0.26 m | 4.36 ± 0.08 gh | 6.97 ± 0.12 k |
| V × D2 × H1 | 14.39 ± 0.31 efg | 109.89 ± 0.11 g | 6.71 ± 0.08 e | 21.48 ± 0.25 g |
| V × D3 × H1 | 11.39 ± 0.31 i | 76.47 ± 0.32 m | 5.34 ± 0.13 f | 41.17 ± 1.01 c |
| T × D1 × H2 | 17.50 ± 0.31 b | 102.96 ± 0.27 i | 9.55 ± 0.13 b | 15.29 ± 0.20 hi |
| T × D2 × H2 | 20.61 ± 0.31 a | 155.41 ± 0.35 d | 10.52 ± 0.23 a | 33.67 ± 0.72 d |
| T × D3 × H2 | 16.39 ± 0.31 bcd | 6.65 ± 0.61 f | 6.64 ± 0.02 e | 11.00 ± 0.18 a |
| P × D1 × H2 | 16.44 ± 0.29 bcd | 107.34 ± 0.33 h | 7.83 ± 0.06 d | 12.54 ± 0.09 j |
| P × D2 × H2 | 19.55 ± 0.29 a | 145.41 ± 0.21 e | 6.50 ± 0.03 e | 20.81 ± 0.08 g |
| P × D3 × H2 | 15.55 ± 0.29 cde | 82.79 ± 0.13 l | 5.34 ± 0.03 f | 41.18 ± 0.23 c |
| V × D1 × H2 | 13.39 ± 0.31 gh | 88.58 ± 0.27 k | 6.77 ± 0.13 e | 10.82 ± 0.20 j |
| V × D2 × H2 | 15.44 ± 0.29 cdef | 170.11 ± 0.39 b | 8.53 ± 0.11 c | 27.28 ± 0.35 f |
| V × D3 × H2 | 16.61 ± 0.31 bc | 185.02 ± 0.03 a | 6.64 ± 0.08 e | 51.10 ± 0.58 a |

Values associated with different letters are significantly different according to Tukey's test at $p < 0.05$. Experimental factors: three quinoa cultivars (Titicaca—T, Puno—P, and Vikinga—V), three crop densities (D1—160 plants·m$^{-2}$, D2—320 plants·m$^{-2}$, and D3—760 plants·m$^{-2}$) and two harvest times (13 November at 52 days after sowing DAS—H1 and 23 November at 62 DAS—H2). DAS—days after sowing.

### 3.2. Phytosterol Contents of Young Quinoa Leaves

Phytosterols are compounds that are found in plants and accumulate especially in juvenile phases of plant growth. Phytosterols have been found to prevent the accumulation of cholesterol in the human body. So far, quantification of phytosterols in young quinoa plants has not been performed. Evidence of these compounds in the juvenile plants could form the basis of an innovative way of promoting consumption of young quinoa plants. Regarding the content of phytosterols in qualitative terms, four compounds were

analyzed in young quinoa plants: ergosterol, stigmasterol, campesterol and β-sitosterol. The influence of the cultivar on the phytosterol content is shown in Table 5. Vikinga had the highest content of ergosterol, Puno had the highest content of stigmasterol and β-sitosterol, while Titicaca had the highest content of campesterol, the differences being statistically significant ($p < 0.05$). The highest total content of phytosterols was recorded by the Puno and the lowest by Titicaca, the differences being significant. Regarding the influence of density on phytosterol compounds (Table 5), at D3 the highest content of ergosterol, stigmasterol and β-sitosterol was obtained, while at D2 the highest content of campesterol was recorded, the differences being statistically significant ($p < 0.05$). Regarding the total phytosterol content, the highest values were obtained at D3, followed by D2 and D1, the differences being statistically significant ($p < 0.05$). The total phytosterol increased at D2 (by 8%) and D3 (by 14%) compared to D1.

**Table 5.** Cultivar, plant density and harvesting time effect on phytosterol compounds of quinoa leaf cultivars.

| Treatment | Ergosterol ($\mu g \cdot 100 \ g^{-1}$ dw) | Stigmasterol ($\mu g \cdot 100 \ g^{-1}$ dw) | β-Sitosterol ($\mu g \cdot 100 \ g^{-1}$ dw) | Campesterol ($\mu g \cdot 100 \ g^{-1}$ dw) | Total Phytosterols ($\mu g \cdot 100 \ g^{-1}$ dw) |
|---|---|---|---|---|---|
| Cultivar | | | | | |
| Titicaca | 1.22 ± 0.01 b | 126.64 ± 0.31 c | 487.71 ± 2.18 b | 4.05 ± 0.02 a | 619.62 ± 2.53 b |
| Puno | 0.90 ± 0.01 c | 136.31 ± 0.27 a | 589.04 ± 2.72 a | 3.12 ± 0.02 b | 729.37 ± 3.02 a |
| Vikinga | 2.39 ± 0.01 a | 131.32 ± 0.36 b | 477.55 ± 2.12 c | 2.72 ± 0.02 c | 613.98 ± 2.50 c |
| Crop density | | | | | |
| D1 160 plants·m$^{-2}$ | 0.51 ± 0.01 c | 119.18 ± 0.34 c | 485.65 ± 2.85 c | 2.99 ± 0.01 b | 608.32 ± 3.21 c |
| D2 320 plants·m$^{-2}$ | 1.58 ± 0.01 b | 132.89 ± 0.36 b | 522.52 ± 2.09 b | 3.93 ± 0.02 a | 660.93 ± 2.48 b |
| D3 760 plants·m$^{-2}$ | 2.42 ± 0.01 a | 142.20 ± 0.25 a | 546.14 ± 2.09 a | 2.97 ± 0.02 b | 693.72 ± 2.36 a |
| Harvest date | * | * | ns | * | * |
| 52 DAS | nd | 139.03 ± 0.25 | 535.45 ± 1.61 | 3.76 ± 0.01 | 678.24 ± 1.88 |
| 62 DAS | 3.00 ± 0.02 | 123.82 ± 0.38 | 500.75 ± 3.07 | 2.84 ± 0.02 | 630.41 ± 3.49 |

Values associated with different letters are significantly different according to Tukey's test at $p < 0.05$. For Harvest time Anova test was performed (* $p < 0.05$, ns—not significant). nd—not detected. dw—dry weight. DAS—days after sowing.

Regarding the effect of harvesting date on the phytosterol compounds (Table 5), the best results were obtained at 52 DAS for stigmasterol, campesterol and total phytosterol content, the differences being statistically significant ($p < 0.05$). Ergosterol was detected only at 62 DAS. No significant differences were observed for β-sitosterol according to the ANOVA test ($p > 0.05$). The total phytosterols decreased by 7% at 62 DAS compared with 52 DAS.

The interaction between cultivar, density and harvesting time is presented in Table 6. Ergosterol was found in very low quantities, from undetectable in many variants up to 7.62 $\mu g \cdot 100 \ g^{-1}$ dw (dry weight) in Vikinga planted at a density of D3 and harvested at 62 DAS. This sterol is mainly found in mature leaves (62 DAS) and at higher planting densities (D3), regardless of cultivar.

Stigmasterol was highest in variant T × D3 × H1, β-sitosterol in variant P × D2 × H1 and campesterol in variant T × D2 × H1, while in variants P × D3 × H2 and V × D1 × H2 it was not detected, the differences being statistically significant ($p < 0.05$). With regards to total phytosterol, variants P × D2 × H1 and P × D2 × H2 had the highest content, the differences being statistically significant ($p < 0.05$).

**Table 6.** Cultivar, plant density and harvesting time interaction effect on phytosterol compounds of quinoa leaf cultivars.

| Interaction | Ergosterol ($\mu g \cdot 100\ g^{-1}$ dw) | Stigmasterol ($\mu g \cdot 100\ g^{-1}$ dw) | β-Sitosterol ($\mu g \cdot 100\ g^{-1}$ dw) | Campesterol ($\mu g \cdot 100\ g^{-1}$ dw) | Total Phytosterols ($\mu g \cdot 100\ g^{-1}$ dw) |
|---|---|---|---|---|---|
| T × D1 × H1 | nd | 160.11 ± 0.18 b | 454.23 ± 1.33 f | 5.11 ± 0.01 b | 619.45 ± 1.51 gh |
| T × D2 × H1 | nd | 161.75 ± 0.43 b | 467.75 ± 4.33 f | 5.25 ± 0.01 a | 634.75 ± 4.78 g |
| T × D3 × H1 | nd | 175.25 ± 0.14 a | 547.37 ± 2.14 d | 4.50 ± 0.03 d | 727.12 ± 2.31 bc |
| P × D1 × H1 | nd | 108.12 ± 0.18 j | 611.50 ± 2.94 ab | 3.12 ± 0.01 j | 722.74 ± 3.14 c |
| P × D2 × H1 | nd | 127.37 ± 0.21 h | 623.12 ± 0.69 a | 3.50 ± 0.03 g | 753.99 ± 0.93 a |
| P × D3 × H1 | nd | 142.00 ± 0.13 e | 598.50 ± 0.29 b | 2.87 ± 0.01 k | 743.37 ± 0.42 ab |
| V × D1 × H1 | nd | 105.62 ± 0.38 k | 457.00 ± 0.58 f | 1.87 ± 0.01 m | 564.49 ± 0.96 i |
| V × D2 × H1 | nd | 132.62 ± 0.36 g | 548.37 ± 2.08 d | 3.37 ± 0.02 h | 684.36 ± 2.46 e |
| V × D3 × H1 | nd | 138.45 ± 0.26 f | 511.20 ± 0.12 e | 4.22 ± 0.01 f | 653.87 ± 0.39 f |
| T × D1 × H2 | 1.54 ± 0.03 g | 79.88 ± 0.51 n | 425.65 ± 3.75 g | 3.44 ± 0.03 gh | 510.51 ± 4.32 k |
| T × D2 × H2 | 2.54 ± 0.03 e | 88.74 ± 0.43 m | 432.14 ± 0.81 g | 3.24 ± 0.02 i | 526.66 ± 1.28 k |
| T × D3 × H2 | 3.25 ± 0.01 d | 94.12 ± 0.19 l | 599.12 ± 0.75 b | 2.75 ± 0.01 l | 699.24 ± 0.97 de |
| P × D1 × H2 | nd | 146.87 ± 0.50 d | 536.75 ± 4.39 d | 4.37 ± 0.02 g | 687.99 ± 4.91 e |
| P × D2 × H2 | 1.75 ± 0.02 f | 147.00 ± 0.31 d | 600.50 ± 3.00 b | 4.87 ± 0.04 c | 754.12 ± 3.37 a |
| P × D3 × H2 | 3.62 ± 0.03 c | 146.50 ± 0.29 d | 563.87 ± 5.02 c | nd | 713.99 ± 5.35 cd |
| V × D1 × H2 | 1.50 ± 0.01 g | 114.50 ± 0.29 i | 428.75 ± 4.10 g | nd | 544.75 ± 4.39 j |
| V × D2 × H2 | 5.20 ± 0.01 b | 139.87 ± 0.40 f | 463.25 ± 1.62 f | 3.37 ± 0.02 | 611.69 ± 2.05 h |
| V × D3 × H2 | 7.62 ± 0.04 a | 156.87 ± 0.46 c | 456.75 ± 4.21 f | 3.50 ± 0.03 h | 624.74 ± 4.74 gh |

Values associated with different letters are significantly different according to Tukey's test at $p < 0.05$. nd—not detected. dw—dry weight. Experimental factors: three quinoa cultivars (Titicaca—T, Puno—P, and Vikinga—V), three crop densities (D1—160 plants·$m^{-2}$, D2—320 plants·$m^{-2}$, and D3—760 plants·$m^{-2}$) and two harvest dates (13 November at 52 days after sowing DAS—H1 and 23 November at 62 DAS—H2). DAS—days after sowing.

### 3.3. Polyphenols Contents of Young Quinoa Leaves

Polyphenols are micronutrients often found in fruits and vegetables with antioxidant activity and with potential roles in preventing cancer, diabetes, cardiovascular and neurodegenerative diseases. In the present study, three compounds were investigated: ferulic acid, isoquercitrin and rutoside.

The influence of cultivar on phenolic content is presented in Table 7. The highest content of ferulic acid was observed in Puno, while Titicaca had the highest content of isoquercitrin and rutoside, the differences being statistically significant ($p < 0.05$). Total phenolic compounds decreased in Vikinga and Puno compared with Titicaca, respectively, the differences being statistically significant.

Regarding the influence of plant density on the phenolic compounds (Table 7), the highest content of ferulic acid and isoquercitrin was obtained at D3, the differences being statistically significant ($p < 0.05$). No significant differences were obtained in the case of rutoside. Regarding the total phenolic compounds, the highest values were obtained at D3, the differences being statistically significant ($p < 0.05$). The total phenolic compounds decreased by 5.8% and 5.2% at D2 and D1 compared with D1.

Regarding the effect of the harvest date on the phenolic compounds (Table 7), the best results were obtained at 52 DAS for ferulic acid, rutoside and total phenolic compounds, the differences being statistically significant ($p < 0.05$). For isoquercitrin, the differences were not significant according to the ANOVA test ($p > 0.05$). The total phenolic compound decreased by 21.4% at 62 DAS compared to 52 DAS.

**Table 7.** Cultivar, plant density and harvesting time effect on phenolic compounds in quinoa leaf cultivars.

| Treatment | Ferulic Acid ($\mu$g·g$^{-1}$ dw) | Isoquercitrin ($\mu$g·g$^{-1}$ dw) | Rutoside ($\mu$g·g$^{-1}$ dw) | Total Phenolics ($\mu$g·g$^{-1}$ dw) |
|---|---|---|---|---|
| **Cultivar** | | | | |
| Titicaca | 41.19 ± 0.21 c | 85.91 ± 0.63 a | 116.55 ± 1.26 a | 243.64 ± 2.10 a |
| Puno | 51.71 ± 0.25 a | 82.35 ± 0.70 b | 48.40 ± 0.73 c | 182.45 ± 1.68 c |
| Vikinga | 47.50 ± 0.20 b | 79.14 ± 0.64 c | 73.23 ± 1.10 b | 199.87 ± 1.94 b |
| **Crop density** | | | | |
| D1 160 plants·m$^{-2}$ | 46.46 ± 0.21 b | 79.47 ± 0.61 b | 79.42 ± 1.05 ns | 205.35 ± 1.86 b |
| D2 320 plants·m$^{-2}$ | 45.39 ± 0.23 c | 79.14 ± 0.61 b | 79.46 ± 1.02 ns | 204.00 ± 1.86 b |
| D3 760 plants·m$^{-2}$ | 48.54 ± 0.22 a | 88.79 ± 0.76 a | 79.29 ± 1.03 ns | 216.62 ± 2.00 a |
| **Harvest date** | * | ns | * | * |
| 52 DAS | 52.08 ± 0.22 | 84.73 ± 0.64 | 96.89 ± 1.16 | 233.70 ± 2.02 |
| 62 DAS | 41.51 ± 0.22 | 80.20 ± 0.68 | 61.89 ± 0.90 | 183.61 ± 1.80 |

Values associated with different letters are significantly different according to Tukey's test at $p < 0.05$. For Harvest time ANOVA test was performed (* $p < 0.05$, ns—not significant). dw—dry weight. DAS—days after sowing.

The interaction between cultivar, density and harvest time is presented in Table 8. Among the compounds analyzed, the lowest content was obtained in the case of ferulic acid (38 $\mu$g·g$^{-1}$ dw) and the highest in the case of rutoside (162.75 $\mu$g·g$^{-1}$ dw). Ferulic acid had the best results in variant P × D3 × H1, isoquercitrin in variants T × D3 × H1 and P × D3 × H2, and rutoside in control T × D1 × H1, and variants T × D2 × H1 and T × D3 × H1, the differences being statistically significant ($p < 0.05$). With regard to the total phenolic compounds, variant T × D3 × H1 had the best result, and variant V × D1 × H2 had the lowest result, the differences being statistically significant ($p < 0.05$).

**Table 8.** Cultivar, plant density and harvesting time interaction effect on phenolic compounds of quinoa leaf cultivars.

| Interaction | Ferulic Acid ($\mu$g·g$^{-1}$ dw) | Isoquercitrin ($\mu$g·g$^{-1}$ dw) | Rutoside ($\mu$g·g$^{-1}$ dw) | Total Phenolics ($\mu$g·g$^{-1}$ dw) |
|---|---|---|---|---|
| T × D1 × H1 | 44.37 ± 0.23 d | 84.36 ± 0.62 c | 162.75 ± 1.44 a | 291.48 ± 2.29 b |
| T × D2 × H1 | 44.37 ± 0.21 d | 82.37 ± 0.64 c | 162.75 ± 1.50 a | 289.49 ± 2.36 b |
| T × D3 × H1 | 44.37 ± 0.22 d | 101.62 ± 1.07 a | 162.75 ± 1.56 a | 308.74 ± 2.85 a |
| P × D1 × H1 | 57.00 ± 0.23 b | 82.37 ± 0.57 c | 51.50 ± 0.81 d | 190.87 ± 1.61 f |
| P × D2 × H1 | 44.37 ± 0.21 d | 82.37 ± 0.59 c | 32.87 ± 0.50 e | 159.61 ± 1.30 hi |
| P × D3 × H1 | 63.25 ± 0.14 a | 82.37 ± 0.61 c | 51.50 ± 0.87 d | 197.12 ± 1.62 ef |
| V × D1 × H1 | 57.00 ± 0.25 b | 82.37 ± 0.52 c | 89.24 ± 1.41 b | 228.61 ± 2.18 c |
| V × D2 × H1 | 57.00 ± 0.24 b | 82.37 ± 0.54 c | 88.62 ± 1.39 b | 227.99 ± 2.17 c |
| V × D3 × H1 | 57.00 ± 0.24 b | 82.37 ± 0.55 c | 70.00 ± 0.98 c | 209.37 ± 1.77 d |
| T × D1 × H2 | 38.00 ± 0.19 e | 82.37 ± 0.47 c | 70.00 ± 1.03 c | 190.37 ± 1.69 f |
| T × D2 × H2 | 38.00 ± 0.20 e | 82.37 ± 0.48 c | 71.02 ± 1.03 c | 191.39 ± 1.71 f |
| T × D3 × H2 | 38.00 ± 0.20 e | 82.37 ± 0.50 c | 70.00 ± 1.01 c | 190.37 ± 1.71 f |
| P × D1 × H2 | 44.37 ± 0.18 d | 82.37 ± 0.69 c | 51.50 ± 0.69 d | 178.24 ± 1.65 g |
| P × D2 × H2 | 50.62 ± 0.35 c | 63.00 ± 0.68 b | 51.50 ± 0.69 d | 165.12 ± 1.72 h |
| P × D3 × H2 | 50.62 ± 0.36 c | 101.62 ± 1.09 a | 51.50 ± 0.75 d | 203.74 ± 2.20 de |
| V × D1 × H2 | 38.00 ± 0.16 e | 63.00 ± 0.78 b | 51.50 ± 0.83 d | 152.50 ± 1.76 i |
| V × D2 × H2 | 38.00 ± 0.16 e | 82.37 ± 0.73 c | 70.00 ± 1.00 c | 190.37 ± 1.89 f |
| V × D3 × H2 | 38.00 ± 0.17 e | 82.37 ± 0.71 c | 70.00 ± 0.99 c | 190.37 ± 1.87 f |

Values associated with different letters are significantly different according to Tukey's test at $p < 0.05$. dw—dry weight. Experimental factors: three quinoa cultivars (Titicaca—T, Puno—P, and Vikinga—V), three crop densities (D1—160 plants·m$^{-2}$, D2—320 plants·m$^{-2}$, and D3—760 plants·m$^{-2}$) and two harvest dates (13 November at 52 days after sowing DAS—H1 and 23 November at 62 DAS—H2). DAS—days after sowing.

## 4. Discussion

In this study the effects of three varieties of quinoa (Titicaca, Puno, and Vikinga) and cultivation technology (density and harvest period) on the production and morphology of young quinoa leaves, and the content of phytosterols and phenolic compounds were evaluated. The first objective of this study was to establish the optimal cultivation technology for obtaining young quinoa plants. From a morphological point of view, a variability was found depending on the variety, so the largest number of leaves was obtained at Puno while the largest leaf area at Vikinga. In terms of mass of fresh plants and production, the highest values were obtained for the Titicaca variety. The variability of these parameters depending on the variety is a characteristic found in many plant species [22].

In order to establish the optimal cultivation technology, respectively the planting density, in this study three crop densities were used 160, 320, and 760 plants·m$^{-2}$. Thus, for the morphological parameters the density of 320 plants·m$^{-2}$ was optimal while for the production the optimal density was 760 plants·m$^{-2}$. Determining the sowing density is an efficient method used to obtain higher yields [23].

In terms of harvest date, all morphological parameters, including production, were higher at 62 DAS. In general, plants tend to accumulate biomass in vegetative organs until the appearance of flowers, after which the synthesis of nutrients is directed to the reproduction [24].

The second objective of the study was to establish the effect of cultivation technology and cultivar on the phytosterol content of young quinoa leaves. Phytosterols (sterols and stanols) are analogous to cholesterol, playing two roles in plants: structural components of cell membranes and precursors in hormone synthesis [25]. To date, over 250 plant sterols are known, the most common being 4-desmethylsterols, including sitosterol, stigmasterol and campesterol [26]. Inclusion of phytosterols in diet has potential beneficial effects on human health, contributing to decreased cholesterol and incidence of cardiovascular diseases[27–29]. Daily intake of phytosterols occurring naturally in the diet can range between 60 mg and 500 mg/day [30]. The phytosterol content of young quinoa leaves is reported for the first time in this study. Four compounds were selected for investigation based on their abundance in plants [26]: ergosterol, stigmasterol, β-sitosterol, and campesterol. All the factors studied (cultivar, planting density, and harvesting date) influenced the content of individual phytosterols as well as total phytosterols. Among the investigated factors, cultivar had the greatest influence on the phytosterol content. β-sitosterol was found in the highest quantity and ergosterol in the lowest quantity in Puno. These results can be explained by the genetic heritability of phytosterol, as was demonstrated by quantitative trait loci (QTL) investigations for phytosterol contents in sunflower and rapeseed [31,32]. According to Nurmi et al. (2010) [33], genotype as well as environmental conditions account for the variation in phytosterol content in wheat between 700 μg·g of dm and 928 μg·g of dm. Similarly, Alignan et al. (2009) [34], reported differences in total sterol and stanol content in wheat up to 30 mg·100 g$^{-1}$ dw due to genetic variation and sowing date. In general, a higher content of phytosterols was obtained in younger plants harvested at 52 DAS compared with 62 DAS. Accordingly, the total phytosterols decreased by 7% at 62 DAS compared with 52 DAS. However, there was an exception in the case of ergasterol, which was found only at 62 DAS. As shown in this study, young quinoa leaves can be an important source of phytosterols, having a content up to 753.99 (μg·100 g$^{-1}$ dw).

The third objective of this study was to highlight the effects of crop management and cultivar on the phenolic content of young quinoa leaves. Plant phenols are secondary metabolic compounds, synthesized from shikimate and phenylpropanoid pathways, that have beneficial effects on human health [35]. Numerous clinical and epidemiological studies have led to the result that polyphenol intake may protect against chronic diseases such as cardiovascular and neurodegenerative diseases, cancer or type 2 diabetes [36]. In this study, three phenols were selected for investigation based on their abundancy in plans and importance for food industry: ferulic acid, isoquercitrin and rutoside. Ferulic acid is a phenolic acid with antioxidant properties; it has potential to be neuroprotective

and enhancing response to heat stress in animals [37,38]. Isoquercitrin is a flavonoid with multiple biological activities including osteogenesis [39]. Rutoside, also called rutin, is a flavonoid glycoside with potential human health benefits in reducing post-thrombotic syndrome, internal bleeding, and hemorrhoids [40,41]. More than 860 products exist on the US market, and 500–2000 mg·day$^{-1}$ oral administration is considered safe [42]. In general, secondary metabolites in plants are protective against stress factors such as ultraviolet radiation and attack by pathogens and pests. Research has shown that the content of these compounds can be increased by various cultivation technologies [43,44]. In this study, the cultivar influenced the phenolic content; the highest content of ferulic acid was recorded in the Puno cultivar, while the Titicaca variety had the highest content of isoquercitrin and rutoside. Variation of phenolic concentration in cultivars has been observed in many species, including basil, sweet pepper, and quinoa [44–46]. Moreover, in this study, the total phenolic content was higher at 52 DAS and the highest density (760 plants·m$^{-2}$). An explanation for the higher synthesis of phenols at higher densities may be correlated with their role in plants to protect against stress, in this case higher density [47]. Quinoa is a super grain that can contribute to the food security both through the seeds it produces but also through its leaves eaten as a salad that have a rich content of nutrients and bioactive compounds [48,49]. The results of this study provide the basis for the selection of cultivars and cultivation techniques for an enhanced content of phytosterols and phenols.

## 5. Conclusions

The highest quinoa production was obtained for the Titicaca variety, at the sowing density of 760 plants·m$^{-2}$ and harvested at 62 DAS. Leaves of quinoa accumulate high amounts of phytosterols and phenols. Higher content of phytosterols in quinoa leaves is present at the first harvest at 52 DAS. Among the phytosterols, β-sitosterol content was the greatest. The crop management of quinoa significantly influences the content of individual phytosterols as well as their total content, both depending on the cultivar, crop density and harvest period. The content of phenols in quinoa is influenced by the cultivar and harvest period. Between polyphenols, rutozid accumulates in the largest quantity at the first round of harvesting at 52 DAS. The results obtained emphasize that the optimum harvesting age is at 52 DAS when the largest amount of phytosterols and phenols have accumulated in the leaves.

**Author Contributions:** Conceptualization, V.S. and V.Z.; methodology, I.M., D.A.-M., A.-M.G. and L.V.; software, M.B. and G.-C.T.; validation, V.S., V.Z., A.-M.G. and G.-C.T.; formal analysis, A.C. and O.-R.R.; investigation, D.A.-M. and O.-R.R.; resources, V.S.; data curation, A.C.; writing—original draft preparation, V.S. and M.V.; writing—review and editing, M.B. and V.Z.; visualization, M.B. and G.-C.T.; supervision, V.S. and V.Z.; project administration, V.S. and M.V.; funding acquisition, V.S. All authors have read and agreed to the published version of the manuscript.

**Funding:** This research received no external funding.

**Data Availability Statement:** Not applicable.

**Acknowledgments:** The authors wish to thank "Ion Ionescu de la Brad" Iasi University of Life Sciences for the financial support of the experiments.

**Conflicts of Interest:** The authors declare no conflict of interest.

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
