# Peer review of "Phytosterol and Polyphenol Contents and Quinoa Leave Yields Variation in Relationships to Variety, Density and Harvesting Date"

_agronomy, doi:10.3390/agronomy12102397_

Round 1

Reviewer 1 Report

Please state clearly what the controls are and associated analysis data. It is confusing as none of the treatments display the same values as the prior tables. 

Author Response

Review 1.

Please state clearly what the controls are and associated analysis data. It is confusing as none of the treatments display the same values as the prior tables. 

Response: Dear reviewer, thank you very much for your comments and suggestions. Below I will respond to the comment. Also, I have included in the corrected version all the suggestions from the pdf.

In the results section there are 6 tables. Tables 3, 5, and 7 show the mean and the standard error of the experience for each variety, for each density and for each harvesting date. Tables 4, 6, and 8 present the mean and standard error of the interactions of the experimental variants.

I improved the introduction based on your suggestions.

In the materials and method, I introduced the reference on the basis of which the technological works were carried out L113-114: 'The cultivation was carried out in accordance with the technological norms described by Stoleru et al., 2014'

Review 2.

Dear Authors,

many thanks for interesting study on variation of phytosterols and phenolic compounds due to cultivars, plant density and harvesting time.

Today, quinoa is interesting not only by its grain, but as well its tolerance to abiotic stresses and the potential use of leaves as green food for human and animal feed.

Title: is suggested and you will find in the pdf version of your manuscript.

Introduction: needs to improve and use suggested bibliographic material that you can find and will provide you more information on similar studies. Suggestions and comments you will find in the pdf version of the manuscript.

Materials and methods: are clear, nevertheless some information are no clear. Please follow the indications in the pdf version of attached document.

Results and Discussions: Please correct the word Punto to "Puno". This section are clear described, and more comments and suggestions to improve you will find in the pdf version.

Conclusions: please, adjust according to the suggested aims from Introduction section.

References: please, add the suggested references mentioned in Introduction section.

Sincerely yours,

The reviewer

Answer: Dear reviewer, thank you very much for your comments and suggestions.

  1. I changed the title to the suggested one.
  2. I have included all bibliographical references in the introduction.
  3. I corrected the materials and methods section according to your suggestions. I included the time of harvesting according to BBCH L105-106: 'Harvesting was done at growth stage 4 according to the BBCH scale (Development of harvestable vegetative parts) before flowering.'

The crop density was changed according to the suggestion from million plants per hectare to plants per square meter.

  1. In the results section I included the moment of harvesting according to the BBCH scale L153-156: 'The number of days until harvest was set so that the plants reached the vegetation stage just before the inflorescence appeared (stage 4 according to the BBCH scale - development of harvestable vegetative parts).'

The titles of the tables have been changed according to the suggestions.

  1. The conclusions section has been adjusted according to the suggestions.
  2. Suggested references have been included.

Reviewer 2 Report

Dear Authors,

many thanks for interesting study on variation of phytosterols and phenolic compounds due to cultivars, plant density and harvesting time.

Today, quinoa is interesting not only by its grain, but as well its tolerance to abiotic stresses and the potential use of leaves as green food for human and animal feed.

Title: is suggested and you will find in the pdf version of your manuscript.

Introduction: needs to improve and use suggested bibliographic material that you can find and will provide you more information on similar studies. Suggestions and comments you will find in the pdf version of the manuscript.

Materials and methods: are clear, nevertheless some information are no clear. Please follow the indications in the pdf version of attached document.

Results and Discussions: Please correct the word Punto to "Puno". This section are clear described, and more comments and suggestions to improve you will find in the pdf version.

Conclusions: please, adjust according to the suggested aims from Introduction section.

References: please, add the suggested references mentioned in Introduction section.

Sincerely yours,

The reviewer

Author Response

(The authors gave the same response as above.)

Round 2

Reviewer 1 Report

the identification of a control group and the comparison to this baseline benchmark data will make the manuscript much clearer. The control group is selected as one variety, at one density and one DAS (e.g., Titicaca, harvested 52 DAS). Everything else is then compared to it. Tab.2 in also confusing: I assume V1 -18 is not "Variants", but replicates.

Author Response

all corrections are do it into the manuscript
